# The Role of Sphingolipids in Regulating Vascular Permeability in Idiopathic Pulmonary Fibrosis

**DOI:** 10.3390/biomedicines11061728

**Published:** 2023-06-16

**Authors:** Girish Jayant, Stephen Kuperberg, Kaumudi Somnay, Raj Wadgaonkar

**Affiliations:** 1SUNY Downstate College of Medicine, Brooklyn, NY 11203, USA; girish.jayant@downstate.edu; 2NYU Grossman School of Medicine, New York, NY 10016, USA; stephenkuperbergmd@gmail.com; 3NY Presbyterian Hospital Queens, New York, NY 11355, USA; ksomnay@gmail.com

**Keywords:** idiopathic pulmonary fibrosis, sphingolipids, sphingosine-1-phosphate, vascular permeability, endothelial barrier dysfunction

## Abstract

Idiopathic pulmonary fibrosis (IPF) is a disease that causes scarring and fibrotic transformation of the lung parenchyma, resulting in the progressive loss of respiratory function and, often, death. Current treatments that target profibrotic factors can slow the rate of progression but are unable to ultimately stop it. In the past decade, many studies have shown that increased vascular permeability may be both a predictive and perpetuating factor in fibrogenesis. Consequently, there is a search for therapeutic targets to try and modulate vascular permeability in fibrotic lungs. One such class of targets that show great promise is sphingolipids. Sphingolipids are common in cell membranes and are increasingly recognized as critical to many cell signaling pathways, including those that affect the integrity of the vascular endothelial barrier. In this focused review we look at sphingolipids, particularly the sphingosine-1-phosphate (S1P) axis and its effects on vascular permeability, and how those effects may affect the pathogenesis of IPF. We further examine existing S1P modulators and their potential efficacy as therapeutics for IPF.

## 1. Introduction

Idiopathic pulmonary fibrosis (IPF) is a devastating disease, characterized by chronic scarring and fibrotic transformation of the lung parenchyma. It has a global prevalence of about 0.33–4.51 cases per 10,000 people after adjusting for age, sex, and smoking status [1]. Patients with this condition suffer from debilitating dyspnea and cough, which is accompanied by a progressive decline in lung function, respiratory failure, and frequently, death. A meta-analysis of six global studies of IPF showed the three-year and five-year cumulative survival rates to be 61.8% and 45.6%, respectively [2]. These figures represent small improvements over past decades thanks in large part to current antifibrotic treatments. The two most important drugs in this category are pirfenidone and nintedanib.

Pirfenidone attenuates fibrosis by downregulating compounds such as transforming growth factor beta (*TGF-β*) and has reduced the rate of lung function decline for IPF patients in three phase III clinical trials [3,4,5,6]. Nintedanib is a tyrosine kinase inhibitor for receptors such as fibroblast growth factor receptor (FGFR) and platelet-derived growth factor receptor (PDGFR). Like pirfenidone, it too has shown reductions in the rate of disease progression [4,7]. However, neither of these drugs has a major effect on mortality or is able to stop progression entirely; they also have many tolerability issues such as potent nausea and rash [4]. Thus, there is a need for the continued study of other pathways which influence the pathogenesis of IPF and therapeutics that modulate those pathways. Most previous research on IPF pathogenesis has focused on fibroblast activation and the epithelial-to-mesenchymal transition of the alveolar epithelium [8,9,10]. More recently, there has been an increase in investigations into the role of vascular permeability in IPF pathogenesis. Increased vascular leak has been associated with fibrogenesis and areas of fibrosis have even been shown to colocalize with areas of increased capillary permeability in the lung [11,12]. This increased permeability may lead to greater extravasation of profibrotic and prothrombotic factors from blood vessels into the alveolar space, sustaining the process of fibrogenesis in fibrotic lungs. As a consequence, examining pathways involved in vascular leakage and seeking ways to regulate those pathways can be the key to finding therapeutics that actually stall disease progression.

Sphingolipids present a myriad of potential targets in this area. Sphingolipids are a class of nearly ubiquitous fatty acid derivatives of sphingosine that influence a variety of cell signaling pathways. Their numerous interactions with *TGF-β* and other profibrotic pathways have made them a subject of interest in the study of IPF pathogenesis [13,14,15]. Compounding research over the past decade shows that they also have numerous effects on vascular permeability. For example, in inflammatory conditions such as sepsis, ceramides can increase vascular permeability while S1P can promote endothelial barrier integrity and reduce vascular leakage [16,17]. This rule appears to apply in the lung vasculature as well. A series of papers have shown that the SphK/S1P axis, especially through S1P receptor 1 (S1PR1), can reduce vascular permeability in the setting of pulmonary fibrosis, and thus, can attenuate the development of IPF [18,19,20]. In this focused review we examine the current research around sphingolipids and the Sphk/S1P axis, and analyze how they affect endothelial barrier integrity, vascular permeability, and the development of IPF. We also discuss the possible benefits and drawbacks of trying to modulate this pathway to control vascular leakage in IPF patients and the role existing S1P modulators could play as potential therapeutics for IPF.

## 2. Models of Pathogenesis of Idiopathic Pulmonary Fibrosis

Idiopathic pulmonary fibrosis (IPF) occurs as a result of irreversible lung scarring that subsequently leads to a decline in lung function. In the popularly held model of pathogenesis, the process starts with injury to the alveolar epithelium. This is the basis for using bleomycin or other compounds with cellular toxicity effects to induce fibrogenesis in murine models [21]. In response to this injury, crosstalk between the epithelium and the extracellular matrix helps repair the damage by activating signaling molecules present in the ECM such as *TGF-β*. These molecules modulate repair by recruiting fibroblasts, activating myofibroblasts, and promoting the transformation of epithelial tissue into mesenchymal tissue [8,10]. The activated fibroblasts continue to secrete collagen, which undergoes abnormal remodeling and crosslinking. Mechano-sensing epithelial cells detect the change in lung stiffness and structure [9].

This change in the mechanical structure of the ECM triggers further myofibroblast proliferation and collagen secretion, creating a positive feedback loop of fibrosis (or a feed-forward cycle, as it has been dubbed in the literature) [10,22]. However, while research has traditionally focused on the interactions between the alveolar epithelium and the ECM in IPF pathogenesis, attention has shifted in recent years to understanding the role that the vascular endothelium plays in this process.

In acute lung injury (ALI), increased vascular permeability is an essential component of the inflammatory response, allowing for the extravasation of immune cells into the ECM. This process is usually short-lived. However, a subject of current investigation is whether this permeability persists as tissue repair begins. As described above, attempts at tissue repair after an acute injury will often result in fibrosis. Notably, studies have shown that the overexpression of pathways such as the Wnt/ β-catenin pathway, which promotes endothelial barrier integrity, can attenuate the development of fibrosis in the days after ALI, implying that continued vascular permeability contributes to fibrogenesis [23], and may thus be applicable to idiopathic pulmonary fibrosis.

For example, increases in vascular permeability are correlated with the formation of fibrotic foci and worse morbidity and mortality outcomes in IPF patients [11,12,22,24]. In particular, McKeown et al. found that higher indexes of permeability in IPF patients were associated with greater rates of rapidly declining lung function and death. Thus, vascular leakage into the alveolar airspace could be a key sustaining factor of fibrogenesis in IPF patients, and therapeutics that target this leakage could have major benefits for disease survivability [11,12]. We discuss one potential model for how this permeability could contribute to fibrogenesis in a later section.

## 3. The Endothelial Barrier

To understand vascular permeability in lung disease processes, we must first understand the endothelial layer. The vascular endothelial layer forms a barrier between the lumen of the blood vessel and the surrounding interstitial tissue. The integrity of this barrier plays a key role in regulating the flow of fluid and macromolecules between the vascular and interstitial space, and transport through this layer must be carefully controlled. Broadly, there are two routes through the endothelium: transcellular and paracellular transport [25]. Transcellular transport is largely mediated by caveolae, which are membrane invaginations that can envelop bloodborne substances in a vesicle and transport them across the cell, releasing them out the other side [26]. Some endothelial beds also contain fenestrae, which are pores that run directly through the cell and allow the transport of materials [22,26]. Paracellular transport, on the other hand, is dependent on the integrity of the inter-endothelial junctions. Tight junctions and adherens junctions both use homophilic adhesion domains to form strong connections between adjacent endothelial cells [27]. Adherens junctions are composed primarily of vascular endothelial cadherins (VE-cadherin); this protein links them to the actin cytoskeleton, allowing for both strength and flexibility in AJ junctions [27]. Their main function is the forming of strong connections between the endothelial barrier cells [28]. Tight junctions, in contrast, are mostly composed of claudins and occludins and they work as a barrier on the apical side of the endothelium, regulating the passage of ions, water, and macromolecules along this paracellular route [27,28].

At rest, the endothelial barrier stringently prevents plasma protein leakage and extravasation of leukocytes and other cells. The endothelial cells actually sequester the proteins needed to interact with leukocytes. They also produce a basal level of nitric oxide (NO) that serves two major functions: maintaining some degree of vasodilation and inhibiting proinflammatory gene expression [29]. In contrast, during inflammatory states, the endothelium allows for leakage of plasma into the interstitial space and increases the recruitment of neutrophils. There are two ways that the endothelium responds to acute inflammation. Type I activation proceeds quickly and without the need for new gene expression, as increased actin filaments connecting to the adherens and tight junctions open up gaps between the endothelial cells, allowing fluid and leukocytes to pass paracellularly. Type 2 activation increases the gene expression of proinflammatory cytokines. For example, tumor necrosis factor alpha (TNF-α) induces further changes in actin cytoskeletons and promotes the internalization of VE-cadherins, sustaining paracellular gaps and endothelial permeability [17,26]. Chronic inflammation begins to involve apoptosis in angiogenesis and vascular remodeling. Of note, in the endothelial lining of the blood–brain barrier, sphingolipids notably play a role in moderating pro- and antiapoptotic signals in this process [17]. As a result, these vessels can develop lasting permeability or damage to the endothelial barrier [17,30,31].

## 4. The Lung Endothelium in Inflammatory States

While the characteristics of the endothelium described in the last section can be applied broadly, there are distinct differences between endothelial tissue in different parts of the body, and these differences could affect potential therapeutic targets for pathologies. The lungs feature a mostly continuous endothelium, lacking the fenestrae that might be seen in the kidneys or other similar tissues [26]. VE-cadherin is particularly important in maintaining strong cell-to-cell adhesion in lung endothelial tissue as this tissue is under constant mechanical stress from respiration [32].

In the lungs, endothelial cells and alveolar epithelial cells are directly connected on their basal sides by a thin basement membrane, allowing for easy gas exchange. However, this means that the endothelial cells are relatively exposed to the external environment, and thus, would need to specialize in having a rapid immune and inflammatory response [26,33]. RNA sequencing analysis of endothelial cells in different tissues confirms that lung endothelial cells have a high expression of genes related to immune system processes such as leukocyte adhesion and trans-endothelial migration relative to other tissues [33].

The expression of adhesion molecules such as intracellular adhesion molecule 1 (ICAM-1) or vascular cell adhesion molecule 1 (VCAM-1) is upregulated in inflammatory states. The binding of neutrophils to these molecules in turn activates many pathways that promote increased vascular remodeling and permeability [32]. In principle, vascular permeability is increased in two different ways. First, inhibiting VE-cadherin functionality can increase barrier permeability [32]. Pathways such as PIK3/AKT pathways and factors such as vascular endothelial growth factor (VEGF) are highly upregulated in lung inflammatory states [34,35,36]. In addition to stimulating angiogenesis, VEGF promotes the dissociation of membrane-bound proteins from cadherins, thus weakening cell–cell junctions and endothelial barrier integrity [37,38]. The second way to increase permeability is through increased actinomyosin contractility: the formation of actin stress fibers causes the contraction of endothelial cells. This opens up paracellular gaps between the cells and increases vascular leakage [32,39]. This second method is of importance to this paper as two pathways in our scope of analysis affect endothelial barrier integrity through this mechanism: the Rho kinase pathway and the sphingosine-1-phosphate pathway.

## 5. Rho Kinases

Rho kinases (ROCKs) are important components in regulating the vascular endothelium response to injury or inflammation. Murine research shows that ROCKs are involved in fibrotic pathways in many different organ systems including the liver, lung, and even kidneys [40,41,42]. There are two isoforms, ROCK 1 and 2. ROCKs are effector molecules of GTPases that mediate the contraction of actin stress fibers in a calcium-independent manner [43]. ROCKs can phosphorylate and activate the myosin light-chain kinase (MLCK), in turn increasing actin cytoskeleton contraction. This causes the cytoplasm of the endothelial cell itself to contract and the cell to “shrink”, thus increasing paracellular gaps between the cells and disrupting the integrity of the endothelial monolayer [44]. There is also the Rac pathway which antagonizes Rho and the ROCKs. Rho increases actomyosin contractility in the cytoplasm, disrupting endothelial paracellular barriers in the manner described above. Rac, on the other hand, counteracts the actions of Rho by modifying the actin cytoskeleton in a calcium-dependent manner and stabilizing the inter-endothelial junctions, reducing vascular permeability [17,22,44]. Many proinflammatory and profibrotic mediators such as VEGF and *TGF-β* promote the Rho/ROCK pathways, increasing vascular permeability as part of their inflammatory processes [45,46]. In contrast, sphingolipids such as S1PR1 can couple with Rac to enhance the endothelial barrier (Figure 1) [17,22,47]. Additionally, ROCK1 and ROCK2 insufficiency in murine models has shown to be protective against bleomycin-induced pulmonary fibrosis through multiple mechanisms including via reduced vascular permeability [41]. Accordingly, they can play an important part in understanding the interplay between sphingolipids and vascular permeability in inflammatory states and fibrosis.

## 6. Sphingolipids

Sphingolipids are a class of fatty acid derivatives of sphingosine that are present almost universally across eukaryotic cell membranes. Two particularly important and closely related sphingolipids are S1P and ceramide. The metabolic relationship between S1P and ceramide is best visualized by the sphingosine rheostat, a term first proposed in 1996 to describe the conceptual model of the relationship between these two molecules. In simple terms, it states that while ceramide is involved in apoptotic and growth-inhibiting pathways, S1P upregulates pathways involved in cell growth and inflammation (Figure 2). Ceramide can be converted into sphingosine by the enzyme ceramidase. Sphingosine can then be phosphorylated into sphingosine-1-phosphate by sphingosine kinase [15].

Some of these steps may play a larger role in regulating the whole pathway. Research from our lab has shown that conversion between sphingomyelin and ceramide is rapidly reversible depending on whether the cell is in an inflammatory or quiescent state; as such, this point in the pathway could be important to maintaining equilibrium between different pro- and anti-inflammatory sphingolipids [48,49]. Sphk is even more central to these pathways; it has been appropriately dubbed the “fulcrum” of the sphingosine rheostat as its activity controls the balance between S1P and ceramide levels (Figure 2) [50]. There are two SphK isoforms: Sphk1 (the more predominant one) and Sphk2 [51]. Despite having identical kinase domains, they have different properties, localize to different cellular areas, and likely have different functions. Our lab conducted early research into the differences between Sphk1 and Sphk2 and found that Sphk1 is significantly more involved in the upregulation of S1P [20].

There are five different membrane-bound G-protein-coupled receptors that S1P can bind to and they are widely distributed across different cell lines (S1PR1-5). This review will focus on S1PR1, but we will briefly touch on the other receptors. S1PR2 is expressed in numerous organs and plays important roles in preventing apoptosis and enhancing growth. The activation of S1PR2 may play roles in both the preservation and disruption of endothelial barriers depending on the organ system [13,52,53]. S1PR3 is the second most common receptor on endothelial cells after S1PR1 [54]. It is involved in many functions, but it is especially crucial to vascular development and the regulation of vascular tone [55,56]. It also has immunomodulatory effects, but there is controversy over whether it is pro- or anti-inflammatory [13,52]. S1PR4 is present mostly in hematopoietic tissues and affects lymphocyte signaling, as well as megakaryocyte and platelet activation. In the CNS it influences dendritic cell activation [52,57]. S1PR5 is most heavily expressed in oligodendrocytes and other myelinating cells. Recent studies have shown that the activation of S1PR5 can be protective against conditions such as multiple sclerosis [52,58].

S1PR1 is the most studied of these receptors. It has numerous effects on cell lines throughout the body. Immune functions include the chemotaxis of lymphocytes from lymph nodes and proinflammatory signaling [15,52]. Its function in the CNS is complex but includes promoting remyelination and astrocyte proliferation. There are currently FDA-approved S1P agonists for use in multiple sclerosis, and such drugs could play a major role in reducing sepsis-related encephalopathy [17]. Lastly, S1PR1 is crucial to maintaining the integrity of the endothelial barrier in the vasculature [17,52].

## 7. The Effects of S1PR1 on Vascular Permeability

The link between sphingolipids and endothelial permeability in lung disease has been increasingly studied in the last decade. The idea was pioneered as far back as 2001 when researchers observed that the dose-dependent addition of S1P produced and sustained electrical resistance across layers of endothelial cells, indicating that S1P was strengthening the integrity of the endothelial barrier [59]. Later, studies established links between S1PR1 and the Rac GTPase pathway, revealing that S1PR1 could reorganize peripheral actin rings, attenuating cellular contraction and preventing the formation of paracellular gaps (Figure 1) [60,61,62,63]. Other research has shown that S1P signaling could affect VE-cadherin interactions, thus strengthening endothelial barrier integrity by manipulating inter-endothelial junctions [64,65].

In this review, we previously discussed the potential links between vascular permeability and worsening fibrosis, morbidity, and mortality in IPF. We have also established that sphingolipids and S1P in particular are tied to the regulation of endothelial vascular permeability. Consequently, it becomes crucial to examine whether modulation of sphingolipids such as S1P can affect vascular permeability in IPF, and if it has any effect on disease outcomes. Numerous previous studies have shown that the infusion of S1P into animal (usually murine or canine) models with acute or chronic lung injury reduces vascular leakage or leukocyte infiltration [20,66,67,68]. These studies, however, did not delve into the specifics of how S1P might affect endothelial cells.

A series of papers by Knipe et al., published in 2018 and 2019, studied the role of S1PR1 and ROCK2 in endothelial cells in IPF [41,69]. They induced endothelial-specific deletion of S1PR1 or ROCK2 in murine models and then induced IPF using a bleomycin challenge. By measuring dye extravasation and hydroxyproline levels, they were able to quantify vascular leakage and the development of pulmonary fibrosis, respectively. They found, as hypothesized, that the loss of S1PR1 resulted in increased vascular permeability and fibrosis while the deletion of ROCK2 was protective against these effects [69]. This fits with our existing model that the S1P/S1PR1 axis works in conjunction with the Rac pathway to stabilize cell–cell junctions and actin cytoskeletons, while the ROCK pathways increase actinomyosin contraction and open up paracellular gaps in the endothelial barrier (Figure 1) [22].

In 2020 and 2022, they used the same murine model with bleomycin-induced pulmonary fibrosis and focused on the results of endothelial-specific deletion of S1PR1. While continuing to measure vascular leakage and fibrosis, these papers also performed flow cytometry and measured D-dimer levels in both serum and bronchoalveolar lavage (BAL). The goal was to assess if S1PR1 knockout led to increased extravasation of clotting factors and inflammatory cells into the alveolar airspaces. Indeed, with the increased vascular permeability in S1PR1 knockout mice, researchers also observed increased coagulation factors and inflammatory cells in BAL samples, compared to controls [19]. Furthermore, the 2022 study by Knipe et al. showed that even low-dose bleomycin was enough to cause a significant increase in fibrosis in S1PR1 knockouts. Histologic staining indicated that fibrotic areas co-localized with areas of increased vascular permeability, strengthening the link between these two factors [18].

Most interestingly, the above-mentioned paper noted that while the knockout mice had increased vascular permeability at baseline, they were phenotypically normal with no increased rates of fibrosis or mortality up to 6 months after S1PR1 deletion. This suggests that increased endothelial permeability does not cause fibrosis by itself, but perhaps strongly amplifies the effect of an inflammatory insult on the lung epithelium [18].

One possible way to conceptualize this is to view increased vascular permeability in S1PR1 knockout subjects as an opening for the extravasation of profibrotic and prothrombotic factors (Figure 3). As these factors leave the blood vessels, they enter the ECM and the alveolar airspaces where they can instigate tissue healing and eventual fibrotic transformation. For example, fibrin extravasation into the alveoli can lead to collagen deposition and maturation, potentiating more permanent fibrotic scarring of epithelial tissue [18]. Thrombin, while involved in coagulation, can also play a role in fibrogenesis. Direct inhibition of thrombin with dabigatran has been shown to reduce integrin αvβ6 induction and TGF-β activation, which then correlated with the reduced development of pulmonary fibrosis [70]. Thus, in states of increased vascular permeability, such as in S1PR1 knockout mice, a greater leakage of prothrombotic factors into alveolar spaces can also induce fibrotic responses in alveolar epithelial cells [18,70]. We must note that Knipe et al. only knocked out S1PR1 in endothelial cell lines, and the effects they observed are only relevant to those cells [18]. This is important because S1P is implicated in other pathways that are potentially profibrotic.

For example, Milara et al. found that S1P is increased in patients with IPF and actually promotes the epithelial–mesenchymal transition through potential crosstalk between the TGF-β1 and S1P/SPHK1 axis [71]. Additionally, a study by Huang et al. found that in the alveolar epithelium, deleting Sphk1 and S1P reduces fibrosis by attenuating the Hippo/YAP pathway and decreasing TGF-β and mitochondrial reactive oxygen species activation [14]. Because S1P could have profibrotic effects in epithelial cells and fibroprotective effects in endothelial cells, cell localization is something that should be considered if designing therapeutic targets for fibrosis based on sphingosine.

Finally, there is a small but important amount of research from the literature showing that S1PR1 may not be protective against vascular leakage and IPF development. The overexpression of S1PR1 can result in the receptor being internalized to the endoplasmic reticulum [72,73]. This action is performed with the help of chaperone proteins such as BiP/GRP78, and other research has shown that BiP/GRP78 can disrupt the endothelial barrier by other pathways that promote the disassembly of VE-cadherins and other endothelial cell–cell junction proteins [74]. Thus, not only could the agonism of S1PR1 eventually lead to desensitization to its anti-permeability effects, but it could trigger other pathways that promote vascular permeability. Research by our lab in 2009 showed that in a lipopolysaccharide-induced (LPS) model of lung injury, the addition of S1P within six hours dramatically attenuated tissue damage and vascular leakage; however, the protective effects of S1P were no longer significant after six hours [20]. A 2010 paper by Shea et al. supported these findings when trying to treat bleomycin-induced fibrosis with nonselective S1P1 agonists such as FTY 270 and AUY954. They found that in contrast to the protective effects of short-term exposure to these agonists, long-term exposure to these agents actually increased vascular leakage, fibrosis, and mortality [75]. These results are consistent with our understanding of S1PR1 internalization and emphasize the importance of considering the temporality of treatment when designing any IPF therapeutic based on sphingosine-1-phosphate.

## 8. The Effects of Other S1P Receptors and Sphingolipids on Vascular Permeability

The bulk of the research and this review thus far have mainly focused on S1PR1. However, there are other S1P receptors to consider and other possible regulatory points in the Sphk/S1P axis. First, S1P can bind to five different receptors, and receptors one through three all have potentially different effects on vascular permeability and pulmonary fibrosis. There is very little research into the effects of S1PR2 on these outcomes. A 2007 paper by Sanchez et al. showed that activation of S1PR2 led to the increased activation of ROCK and phosphate and tensin homolog (PTEN) pathways and caused increased vascular permeability, while the blockage of S1PR2 led to decreased permeability [53]. In 2018, news studies showed that S1PR2 might also affect pulmonary fibrosis on a transcriptional level; S1PR2 deletion reduced the expression of profibrotic cytokines such as IL-13 and IL-4 in bleomycin-induced pulmonary fibrosis [76]. Thus, it seems that S1PR2 might play the opposite role as S1PR1 in IPF, but the paucity of research makes any potential therapeutic role in IPF highly unclear.

S1PR3 also seems to have a profibrotic function. Studies have found that S1PR3 can promote epithelial-to-mesenchymal transformation and fibroblast activation in epithelial cells and ECM, respectively [66]. Additionally, in endothelial cells, S1PR3 seems to promote increased endothelial barrier permeability. Murakami et al. found that S1P binding to S1PR3 can induce vasoconstriction and increased capillary permeability by activation of the ROCK pathway [13]. Additionally, Sammani et al. found that S1PR3 knockout mice were protected against endothelial barrier disruption in IPF models [77]. All of this indicates that S1PR3 probably plays an antagonistic role to S1PR1 in regulating endothelial barrier integrity.

Sphk can also play an important role by determining levels of S1P. As described previously, Sphk is the fulcrum of the rheostat, and though it is not as heavily studied in relation to IPF as S1P, it can still be an important regulatory point. Our lab demonstrated in 2009 that the knockout of Sphk1 exacerbated vascular leakage in an LPS-induced model of lung injury. We also found that the reintroduction of Sphk1 to endothelial cells via the adenovirus vector significantly attenuated permeability and lung damage [20]. In contrast, we found that the overexpression of Sphk2 in the Sphk1 knockout mice increased vascular leakage. While Sphk1 upregulates the production of S1P, SPhk2 seems to have a differing function, and this should be considered if trying to use sphingosine kinase as a therapeutic target [20]. This function of Sphk1 in acute and subacute lung injury has largely been supported by reviews of the literature [66]. However, there is conflicting research: Wang et al. showed that the upregulation of Sphk1 increased lung injury, while Sphk1 inhibitors attenuated vascular permeability. More research is needed to reconcile these differences in the literature [78].

## 9. S1P Modulators and Sphingolipid-Based Treatment Options

While numerous S1P agonists and modulators have been tested in the research, there are four major S1P modulators that are currently in clinical use: fingolimod, siponimod, ozanimod, and ponesimod. This review will focus on these four modulators and any potential they have for the treatment of idiopathic pulmonary fibrosis.

Fingolimod (FTY 720) is the oldest and most commonly used of the four. It is a nonselective modulator of S1PR1, -3, -4, and -5. At high concentrations, it can also activate other sphingolipids such as Sphk and ceramide synthase. It was initially believed to be a pure S1PR1 agonist, but studies have shown that depending on concentration and time of exposure, it can also act as a functional antagonist by internalizing S1PR1 receptors [52,72]. The FDA recently approved this drug for the treatment of relapsing multiple sclerosis (MS) as its anti-inflammatory action can prevent immune cell infiltration to the CNS. Numerous phase III trials have shown that this drug has benefits in decreasing relapse rates of MS flare-ups and preserving brain volume, though it still shows little effectiveness in slowing disease progression [79,80,81,82]. Given its agonism of S1PR1, it could be a promising therapeutic for IPF, and studies have shown that it can reduce fibrosis in renal and hepatic tissue [83,84]. However, the demonstrated results in pulmonary fibrosis are mixed and controversial. On the one hand, some studies have shown that FTY720 can attenuate factors that contribute to lung injury and fibrosis [85,86]. One study found that FTY720 reduced the expression of TGF-β and VEGF, for example, after paraquat-induced fibrosis in murine models [85]. This drug has also reduced vascular permeability in vivo and in vitro, which tracks with our previously held model of S1P1 antagonism; in some studies, a single injection of FTY720 was enough to significantly reduce vascular extravasation after acute lung injury [66,68]. However, these antifibrotic effects come with many caveats. While low doses of FTY720 can attenuate fibrosis, high doses can exacerbate it. Muller et al. showed that low doses (up to 0.1 mg/kg) reduced vascular permeability while higher doses (up to 2 mg/kg) broke down the endothelial barrier by inducing endothelial apoptosis [87].

As discussed previously, another inquiry noted that while FTY720 and other S1P agonists can attenuate vascular leakage in the short run, prolonged exposure can increase vascular permeability, fibrosis, and even cause death [75]. Gendron et al. hypothesized that this might be because FTY720 has different effects in the “inflammatory phase” vs. the “remodeling” phase of fibrotic lung injury [88]. Most murine and other in vivo models simulate lung fibrosis by exposing subjects to an insult such as bleomycin; according to Gendron, this insult would initially cause an inflammatory response and then remodeling of the repair tissue damage, and this remodeling phase might be more representative of idiopathic pulmonary fibrosis. They found that the addition of FTY720 during the inflammatory phase reduced fibrosis, but increased fibrosis when added during the remodeling phase. Importantly, they noted that FTY720 was associated with an increase in the expression of connective tissue growth factor in the remodeling phase, but in neither phase was it associated with an increase in vascular permeability [88]. This is in contrast with the findings of Shea et al. and Muller et al.

There are a few explanations to untangle these contradictory pro- and antifibrotic effects. First, prolonged exposure or high-dose exposure to S1P agonists such as FTY720 might cause the internalization of S1PR1 receptors, and thus, may work as functional antagonism [72]. Second, S1PR1 affects numerous pathways, both pro- and antifibrotic. Several of the papers by Knipe et al. studied how the deletion of S1PR1 induced increased vascular permeability and subsequently increased fibrotic transformation, implying that S1PR1 has antifibrotic effects [18,19]. However, they only studied this model in endothelial cell lines. By contrast, in epithelial cells, S1PR1 can be profibrotic. For example, FTY720 can stimulate the expression of profibrotic genes and connective tissue growth factor via the PI3k/AKT pathway in epithelial cells [89]. As a result, while FTY720 might attenuate vascular leakage and fibrosis in the inflammatory phase, it could also promote fibrosis by enhancing profibrotic gene expression in the remodeling phase. Finally, there is the fact that FTY720 is not specific to S1PR1 but rather targets S1PR1, -3, -4, and -5, and might also stimulate S1PR2 to some degree. Hence, its effects on S1PR2 and -3 could counter its S1P1 agonist effects [89,90].

The research on the remaining three modulators and their effects on IPF is far more sparse. They are indicated for the treatment of a number of inflammatory conditions such as MS or ulcerative colitis (UC) [91]. Unlike fingolimod, the other three therapeutic agents do not require phosphorylation to be active [52]. However, a key point is that they are all more specific than fingolimod. Ozanimod is a potent agonist of S1PR1 and -5. Siponimod is a functional antagonist of those two receptors and ponesimod is a selective antagonist of S1PR1 only with rapid reversibility [52]. This specificity could help overcome some shortcomings fingolimod has as a potential IPF treatment. For example, ozanimod does not have S1PR3 agonist activity, and thus, does not promote fibroblast proliferation and differentiation as strongly as fingolimod [91]. A study of the effects of fingolimod and ponesimod on fibrotic transformation in bleomycin-injured lungs found that ponesimod did not induce extracellular matrix synthesis as much as fingolimod, predominantly because it had a lower potency activation of S1PR3. Additionally, ozanimod’s strongest effects seem to be on attenuating an immune response and have been studied for enhancing the efficacy of COVID-19 vaccines [92]. These immunomodulatory effects could also help with attenuating an inflammatory response in acute lung injury, but would not be as helpful in dealing with chronic IPF. Overall, the receptor specificity of ozanimod, ponesimod, and siponimod could make them more reliable treatments of IPF than fingolimod, but far more studies are needed.

Other S1P agonists such as AUY 954 and SEW2871 exist, but are still limited only to basic science and translational research. However, they should be mentioned because they might confer even greater receptor specificity than ozanimod and ponesimod. For example, Sobel et al. found that SEW2871 had even less profibrotic potential than ponesimod [89].

## 10. Angiogenesis in IPF

The models of IPF we have discussed thus far support a direct link between increased angiogenesis, vascular permeability, and fibrotic transformation. However, we must also address discrepancies in the literature surrounding this idea. Some studies of biopsied IPF lung tissue show markedly increased capillary density in fibrotic areas compared to healthy lung, and these new blood vessels are usually characterized by poor barrier integrity and long-lasting increases in vascular permeability [31,32,93]. Research also shows that the increased expression of factors such as VEGF that increase endothelial permeability are associated with increasing severity and progression of IPF [94,95]. This fits with our model of IPF pathogenesis (Figure 3). However, other studies have shown that the most fibrotic foci in IPF lungs have a decreased expression of VEGF and the greatest vascular density is in the nonfibrotic areas of the lung parenchyma [96]. As with many such cases, this contradictory presentation may be explained by heterogeneity between different endothelial cell lines; single-cell RNA sequencing has found that certain endothelial cell lines are more associated with VEGF expression than others in bleomycin-induced fibrosis [97]. An alternative explanation is that increased capillary density and permeability precedes fibrotic transformation; the high capillary density in nonfibrotic foci could actually be a precursor to the eventual fibrotic transformation of those areas (Figure 3). However, the literature is not in full agreement on the timeline and spatial arrangement of this transformation.

## 11. Future Study

Our review of the literature shows the potential importance of vascular permeability in the pathogenesis and progression of IPF and the role that sphingolipids, particularly the S1P/S1PR1 axis, may play in regulating that permeability. S1PR1 may represent an important future therapeutic target for the treatment of IPF, but there are still gaps in the literature that need to be addressed.

First, there is still disagreement in the literature on the exact role of angiogenesis in the pathogenesis of IPF consensus still does not exist regarding the exact role of angiogenesis in the pathogenesis of IPF. We present compelling evidence that increased vascular permeability is associated with the increased development of lung fibrosis [18,19,69]. However, other papers show that vascular density is greatest in the nonfibrotic foci of IPF lungs [96]. This leads to the question: are those areas of increased vascular density compensating for fibrotic damage to gas-exchange units elsewhere in the lung, or are they precursors to eventual fibrotic transformation (Figure 4)?

Secondly, far more research is needed into different S1P modulators. The currently available S1P modulators are exciting drugs with already recognized uses for diseases such as MS or UC. However, they are not ideal for use in pulmonary fibrosis, predominantly because of their lack of specificity. Drugs such as fingolimod are not specific to S1PR1 and can promote increased vascular permeability through other S1P receptors. Therefore, further studies will have to focus on newer modulators such as SEW2871 that have greater receptor specificity [89]. Additionally, there is great heterogeneity in the effects of the S1PR1 agonism. While the S1PR1 agonism in endothelial cells promotes endothelial integrity and attenuates fibrosis, it can stimulate profibrotic pathways in epithelial cells [14,71]. Additionally, the persistent agonism of S1PR1 can lead to receptor internalization and desensitization [75]. Consequently, in order to use any future S1P modulators as IPF therapeutics, research into optimizing exposure time and drug delivery to only endothelial cell lines is necessary (Figure 4).

Finally, research in this field has vastly focused on the S1P/S1PR1 axis. However, there is potential to manipulate this axis by regulating the sphingosine kinase (as the fulcrum of the sphingolipid rheostat) or other sphingolipids.

## 12. Conclusions

A review of the literature reveals mounting evidence that increased vascular permeability may play an important role in the pathogenesis of IPF. A compelling conceptual model has been expressed in some of the literature to help explain this; increased permeability allows extravasation of profibrotic and prothrombotic factors into alveolar airspaces and triggers the transformation to fibrotic tissue [18,22,70]. Consequently, modulating vascular permeability represents a promising therapeutic approach to treating IPF or stopping its progression. Sphingolipids represent one viable way to modulate that permeability. S1P/S1PR1 has been most heavily studied, and stimulation of S1PR1 can promote endothelial barrier integrity via interactions with Rac and other intracellular pathways. A growing body of research shows that S1PR1 can protect against vascular leakage and increased fibrosis in bleomycin models of IPF. In order to make this pathway effective therapeutically, agonists more specific to S1PR1 and more targeted to endothelial cell populations are necessary.

## Figures and Tables

**Figure 1 biomedicines-11-01728-f001:**
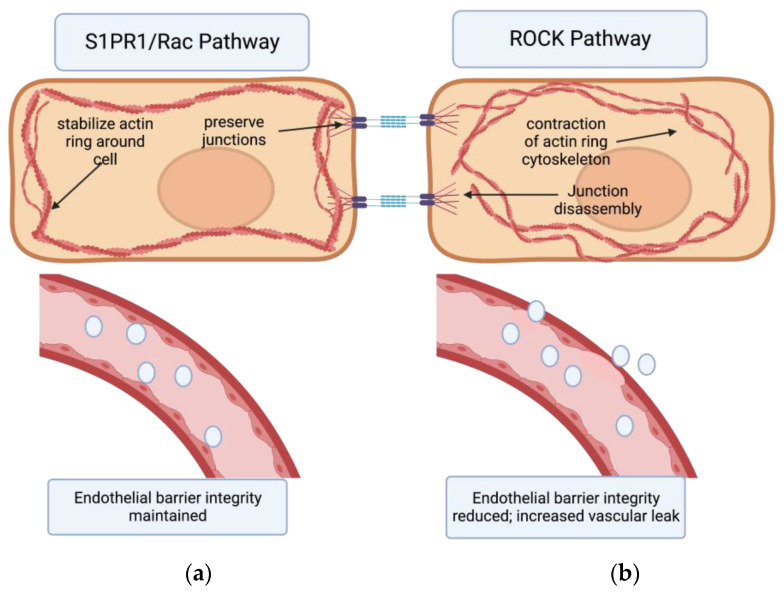
The effects of S1PR1/Rac pathway vs. ROCK pathway on endothelial barrier permeability. (**a**) Binding of the S1PR1 receptor activates the Rac pathway in endothelial cells. This primarily stabilizes the actin cytoskeleton and secondarily preserves endothelial junctions. In turn, endothelial barrier integrity is maintained. (**b**) Activation of the Rho kinase (ROCK) pathway primarily causes formation of actin stress fibers and circular contraction of the actin cytoskeleton, as well as cadherin junction disassembly. This contraction of actin cytoskeleton opens up paracellular gaps between endothelial cells, increasing vascular permeability [17,22,44,47]. Created with BioRender.com (accessed on 21 May 2023).

**Figure 2 biomedicines-11-01728-f002:**
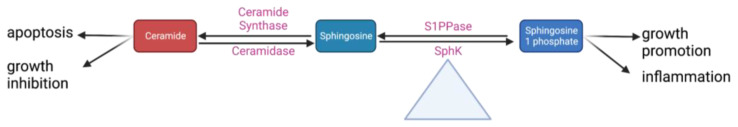
A simplified schematic of the sphingosine rheostat [15,48]. Created with BioRender.com (accessed on 21 May 2023).

**Figure 3 biomedicines-11-01728-f003:**
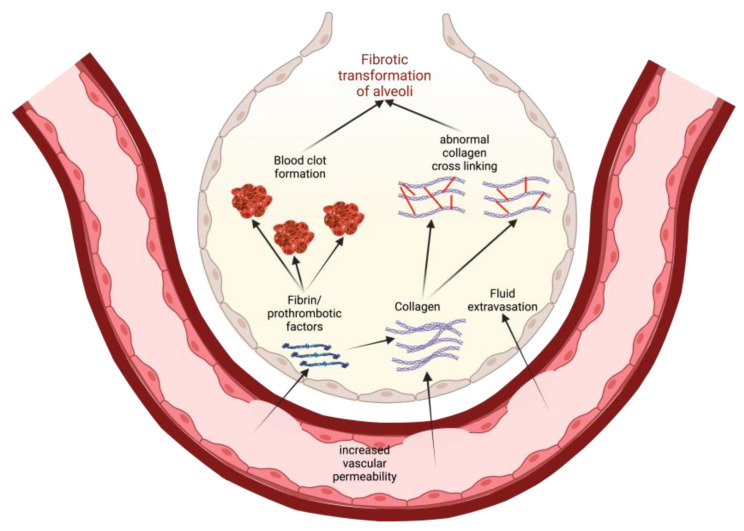
A model for how increased vascular permeability can potentiate fibrotic transformation of alveoli in IPF. This permeability allows for extravasation of prothrombotic factors and collagen that, in turn, promote clotting and activate profibrotic pathways. Eventually, there is recruitment of fibroblasts, abnormal collagen crosslinking, and a general epithelial-to-mesenchymal transformation in the alveoli, leading to lung fibrosis [18,70]. Created with BioRender.com (accessed on 21 May 2023).

**Figure 4 biomedicines-11-01728-f004:**
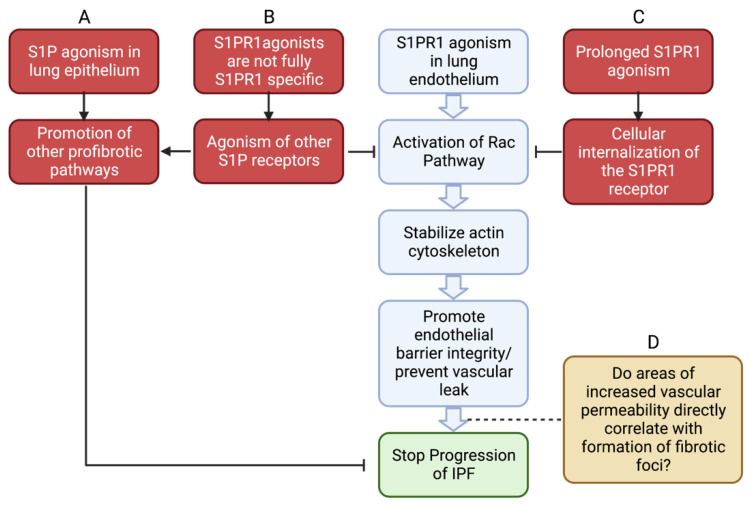
A model for how S1PR1 agonism could prevent progression of IPF and current issues with this therapeutic model, based on a review of the current literature. (**A**) While S1PR1 agonism in lung endothelium can be antifibrotic, S1P agonism in epithelial cells can activate profibrotic pathways. (**B**) Current S1PR1 agonists are not specific and might interact with other S1P receptors, working against the Rac pathway or activating other profibrotic pathways to counter the antifibrotic effects of S1PR1 agonism. (**C**) Prolonged agonism of S1PR1 can cause internalization of those receptors, effectively causing S1PR1 antagonism. (**D**) Our model of reducing vascular leakage to stop IPF progression is widely supported by the literature, but there is still contention about how direct the relationship between areas of vascular leakage and foci of lung fibrosis is [4,18,19,69,71,72,89,90,96]. Created with BioRender.com (accessed on 21 May 2023).

## Data Availability

No new data were created or analyzed in this study. Data sharing is not applicable to this article.

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
