# Peer review of "The Role of Sphingolipids in Regulating Vascular Permeability in Idiopathic Pulmonary Fibrosis"

_biomedicines, 2023, doi:10.3390/biomedicines11061728_

Round 1
Reviewer 1 Report
It is a well-written review article discussing the role of sphingolipid vascular permeability during fibrogenesis. The authors have introduced the basic idea of lung fibrosis, and how endothelium dysfunction was identified during fibrosis. Then the authors discussed the role of sphingolipid, ROCK pathway, S1P and S1PR1 pathway during the process. Most of the concepts have been covered as the title incorporated.
Some minor suggestions for the authors to consider:
1. Can section 4 and 5 combined and make it shorter. If we are discussing the related s1p pathway during vascular permeability during fibrosis, talking about hypoxia might be a bit off-topic. I understand oxygenation is directly correlated with vascular dysregulation, while this section looks suddenly appeared. I would suggest either shorten this section, and focus on inflammation, or explore the hypoxia conditions in the sections discussed about ROCK/S1P/S1PR1 pathway as well. In this way, it could help with more focus of the topic.
2. Is the any possibility for author to provide tables or schematics to conclude the role of different molecules (ROCK/S1P/S1PR1 etc) in vascular permeability dysregulation occurred during fibrogenesis.
Author Response
- As per suggestion, we have combined sections 4 and 5. The section now focuses more on lung endothelium in inflammatory states, specifically the two main mechanisms by which these inflammatory states can increase vascular permeability. And we emphasize how that relates to discussion of ROCK and S1PR1/Rac pathways. Discussion of Hypoxia inducible factor/ hypoxia is deemphasized. Paragraph on distribution of angiogenesis in IPF is moved to its own section later in the article to improve flow of topics.
- We created Figure 1 to show how ROCK and S1PR1 pathways contrast in their effects on actin cytoskeleton and vascular permeability.
Reviewer 2 Report
The Authors describe the role of sphingolipids in Idiopathic Pulmonary Fibrosis. The review is well- and clearly written. It is based on the newest literature in the field. The article contains a comprehensive portion of knowledge that will be important for clinical and basic scientist from the field.
If there is any implication of hormones to sphingolipid regulation it will be nice to have it presented here (relaxin?).
A schematic drawing summarising current knowledge and future unknowns is needed if possible
Author Response
- This didn’t show up much in our review of the literature and we didn’t think discussing what information is available would fit within the flow of the review.
- We created Figure 4 to showcase our model, based on review of the literature, of how S1PR1 agonism can reduce IPF progression. This figure also showcases issues with this potential therapeutic model and thus highlights areas of future research that will be needed if we are to find clinical application for this pathway.